# Layered-Oxide Cathode Materials for Fast-Charging Lithium-Ion Batteries: A Review

**DOI:** 10.3390/molecules28104007

**Published:** 2023-05-10

**Authors:** Xin Meng, Jiale Wang, Le Li

**Affiliations:** Shaanxi Key Laboratory of Industrial Automation, School of Mechanical Engineering, Shaanxi University of Technology, Hanzhong 723001, China

**Keywords:** layered oxide, cathode materials, fast charging, lithium-ion batteries

## Abstract

Layered oxides are considered prospective state-of-the-art cathode materials for fast-charging lithium-ion batteries (LIBs) owning to their economic effectiveness, high energy density, and environmentally friendly nature. Nonetheless, layered oxides experience thermal runaway, capacity decay, and voltage decay during fast charging. This article summarizes various modifications recently implemented in the fast charging of LIB cathode materials, including component improvement, morphology control, ion doping, surface coating, and composite structure. The development direction of layered-oxide cathodes is summarized based on research progress. Further, possible strategies and future development directions of layered-oxide cathodes to improve fast-charging performance are proposed.

## 1. Introduction

Recently, increased energy use has depleted fossil fuels and increased environmental contamination. Accordingly, exploring different green and renewable sources of energy, such as wind energy, waterfalls, solar energy, geothermal, and ocean waves, as well as mitigating environmental contamination, have attracted considerable attention [1,2,3,4,5,6,7]. However, the electricity produced from these green and renewable energy sources depends on nature and is discontinuous, and therefore must be stored for further usages. Devices for storing electrochemical energy including lead-acid batteries, lithium-ion batteries (LIBs), supercapacitors, and electrolyzed water are considered reliable, efficient, and feasible electrical energy storage and conversion methods [8,9,10,11,12,13]. Especially, LIBs with excellent energy and power density and good cyclic stability have been developed and industrialized. Encouraged by the policies of various countries globally, new energy automobiles, with the advantages of energy-saving, cost-effectiveness, and minimal emissions, have developed rapidly, whereby LIBs have gained widespread application for providing power [14,15,16,17]. Unlike consumer electronic products, LIBs for automobiles have more exacting preconditions, including manufacturing costs, power/energy density, security, working life, and mileage anxiety. Cathode materials, which are used as a constituent of LIBs, perform a crucial function for these characteristics [18,19,20,21,22]. In particularl, fast charging of the LIBs used in automobiles is needed to resolve mileage anxiety [23,24].

With respect to cathode materials, layered oxides are some of the most thoroughly explored as they offer high capacity (200–300 mAh g^−1^), exceptional potential (more than 3.5 V vs. Li^+^/Li), and high cost-effectiveness (less or no use of Co) [23]. Therefore, layered oxides are prospective candidates in the design and development of state-of-the-art fast-charging LIBs [25,26,27,28,29]. The desire for perfect fast-charging cathode materials, on the other hand, has always created a tradeoff that has presented a significant problem. (1) Most layered oxides have low initial coulombic efficiency (ICE) in the course of the first charge/discharge process. (2) The discharge voltage of a layered oxide decreases continuously during a cycle, resulting in the loss of energy density. (3) An additional technical problem in ensuring an extended working life for layered oxides is the substantial capacity degradation that occurs over time. (4) Layered oxides can suffer from poor rate performance due to slow reaction kinetics and low electrical conductivity, resulting in poor rate performance. (5) Transition metal dissolution can occur. Layered oxides are severely reduced due to lattice distortion and Mn^2+^ dissolution, as well as lattice distortion resulting from the Jahn–Teller effect [24,30,31,32].

Thus far, overcoming the above technical obstacles and improving the fast-charging performance of layered oxides is still a challenge. Numerous ground-breaking techniques focusing on various Li^+^ insertion cathode materials have been utilized for the development of fast-charging layered oxides. Over the past few years, researchers have explored various strategies to optimize the fast-charging ability of layered oxides. Since the basic understanding, as well as the reaction mechanism of these enhancement approaches, are not completely elucidated, the strategy of designing high-performance fast-charging layered oxides should be given priority, and attention should be paid to its broad application prospects. This review describes the unique electrochemical characteristics of layered oxides. A detailed discussion of the mitigation of the technical challenges of layered oxide fast charging is reported. Finally, the application prospects of layered oxides in fast-charging LIBs are proposed.

## 2. LIB Fast Charging Principles

The superlative LIB has prolonged life, high energy, and power density that can work for a long time on a single charge and can be charged rapidly under any condition. Unfortunately, the physical nature of these requirements leads to trade-offs [33]. For instance, electrodes with great thickness require higher energy density and are affected more severely by concentration and potential gradients created by fast charging [34,35]. As the ambient temperature decreases, the charging rate and recommended maximum voltage are usually maintained at a low level to enhance safety and overall performance, making the working temperature a major obstacle to fast charging [33]. During the charging process, the risk of lithium electroplating increases considerably as the temperature continues to decrease, which affects the retention capacity. The temperature thresholds for lithium electroplating rely on a number of variables, such as the battery characteristics, C rate, and service life [36].

LIBs generally include a graphite anode, a layered spinel or olivine structured lithium metal oxide cathode, a porous polymer separator, a copper/aluminum current collector, and a liquid electrolyte that contains a blend of salts, organic carbonates, salts, and additives. Inside the LIB, whether inside the electrode or at the key interface, achieving reliable operation and fast charging is important [18], which depends on various parameters, including temperature and ion transmission [33]. As illustrated in Figure 1, the charging of LIBs leads to the transfer of ions from the cathode to the anode through the electrolyte. The critical processes controlling this process include ion transfer (1) via the solid electrode, (2) via the electrolyte/electrode interactions of cathode and anode, and (3) via the electrolyte, such as Li^+^ desolvation and solvation. In optimum situations, it would be advantageous for these main phenomena to involve battery charging [36,37]. In general, a cathode comprises an active material, a conductive carbon additive to make enough electronic conductivity, a binder, which takes a vital role in holding the structural integrity of the electrode, and an Al current collector playing the part of a substrate for the electrode coating and as an electronic conductor to the cell terminal. For layered-oxide cathodes, delithiation occurs via Li^+^ de-insertion and accompanying electron transport. Therefore, two types of carriers have their paths, an electron path and the Li^+^ ion transport path [38]. During battery charging, electrons move further from the active material particles (which may involve conductive carbon particles) to the aluminum collector and through the external circuit to the anode, thereby experiencing ohmic resistance [38,39]. The electron path is reversed during the discharge. In both directions, the electron transport should meet the same electronic resistance, and the Li^+^ transfer path dynamics in the composite electrode indicate differences in the charging and discharging process. During the battery charging process, Li^+^ ions move into the electrolyte medium through a particle surface layer and the electrode/electrolyte interface in the active material particles and dissolve before reaching the anode. The discharge path (lithiation) is the opposite but includes the desolvation process before Li+ ions enter the active material [38,40]. In an ideal situation, it would be advantageous for these phenomena to involve battery charging. However, the operating conditions of a battery may cause a series of side effects that affect its performance and life. Moreover, the thermal properties of the battery depend on various circumstances to a great extent. A high charge and discharge current, concentration polarization, or battery resistance accelerates the rate of heat production and has an adverse impact on the safety and performance of the process.

The high-efficiency performance of the energy and power density, kinetics and thermodynamics of the electrochemical reaction and the rate-capable library activity are subject to a slow charge diffusion and transport process of the solid electrode material. Therefore, selecting suitable candidate materials and reasonable material design to improve the diffusion kinetics of Li^+^ and e^−^ in the electrode materials is crucial in promoting fast charging of LIBs.

## 3. Application of Layered Oxides in Fast-Charging LIBs

The lithium ionization/delithiation process of most cathode materials is reliant on the insertion mechanism in which lithium ions are inserted and removed from the plane or tunnel of the host structure. The layered oxides discussed here mainly include spinel LiNi_0.5_Mn_1.5_O_4_ (LNMO), LiNi_x_Co_y_Mn_z_O_2_ (NCM), LiNi_x_Co_y_Al_z_O_2_ (NCA), and Li-rich Mn-based (LRM). The limited capacity of the above cathode materials is caused by the limited number of lithium ions (up to one lithium ion per M). Their speed capacity mainly depends on their electronic and ion conductivity, and the relevant dimensions of lithium-ion transport in their crystal structure.

### 3.1. Spinel LiNi_0.5_Mn_1.5_O_4_ (LNMO)

The spinel LNMO is an extremely attractive cathode material that promotes fast-charging LIBs due to its unique three-dimensional path that can quickly transport lithium ions quickly in all directions and high energy density because of its theoretical capacity (147 mAh g^−1^) and high-working voltage (4.7 V vs. Li) [41,42]. Nonetheless, during the high-pressure fast-charging operation of LNMO, the integration of parasitic reactions and electrolyte oxidation lead to rapid capacity degradation, particularly with the high temperatures generated during the fast-charging process, which leads to the dissolution of manganese and reduces cycle life [41,43]. Moreover, the inherent lithium-ion diffusivity and electronic conductivity of LNMO are low, thereby limiting its fast-charging performance [40]. Therefore, the electrode structure, doping method, and surface coating are very important for realizing the fast-charge transfer kinetics and low-resistance path required for fast-charging LIBs.

Element doping is a facile method for modifying the fast-charging performance of LMNOs. In 2019, Liu et al. developed a cation/anion (Na/F) co-doping method to enhance the electrochemical properties of lithium-rich manganese-based layered-oxide Li_1.2_Ni_0.2_Mn_0.6_O_2_ (Figure 2a,b) [44]. A nitrogen-doped sample delivered good cycle performance due to the enhanced structural stability, whereas a F-doped sample exhibited good capacity and excellent rate performance, primarily because of the improved electronic and ionic conductivity. As a result, the co-doped sample Li_1.12_Na_0.08_Ni_0.2_Mn_0.6_O_1.95_F_0.05_ exhibited good cycle performance (100% following the completion of 100 cycles at 0.2 C) as well as excellent capacity rate (167 mAh g^−1^ at 5 C) (Figure 2c). Moreover, this remarkable improvement was examined systematically using high-resolution transmission electron microscopy, including the Raman spectroscopy and the electrochemical impedance spectroscopy, proving that Na-doping can more effectually prevent the phase change from the layered structure to the spinel-like structure. In addition, the F-doping technique may promote the performance of ionic and electronic conductance. Wei et al. systematically investigated how boron doping influences the morphology, crystal structure, and electrochemical properties of LNMO materials (Figure 2d,e) [45]. Structural analysis showed that the particle sizes, as well as lattice constants of the four boron-doped LNMO-B materials, increased mildly as boron doping continued to increase. Further, boron doping induces the generation of an increased concentration of Mn^3+^ ions and enhances the stability of the structure. The electrochemical test results exhibited how the electrochemical performance of the four boron-doped LNMO-B samples was enhanced in contrast with pure LNMO. With respect to rate performance, the boron-doped LNMO-B (LNMO-B0.01) cathode showed an optimizing rate performance as demonstrated by its discharge capacities of 123.1 mA h g^−1^ at 3 C, 114.5 mA h g^−1^ at 5 C, 104.5 mA h g^−1^ at 7 C, and 82.9 mA h g^−1^ at 10 C, respectively. Conversely, pure LNMO has been shown to gain a substantially minimal discharge capacity of 111.7 mA h g^−1^ at 3 C, 89.1 mA h g^−1^ at 5 C, 61.5 mA h g^−1^ at 7 C, and 10.1 mA h g^−1^ at 10 C, respectively (Figure 2f). The capacity retention of 83.3% following 500 cycles at 3 C was observed in LNMO-B0.01, indicating that it showed the optimum cycling stability. A significant portion of the enhanced electrochemical characteristics of LNMO-B0.01 was attributed to the appropriate quantity of boron doping, which increases the sizes of the particles and reduces the amount of Li_x_Ni_1−x_O impurities. Further, boron doping improves structural stability due to the strong boron–oxygen bond, higher Mn^3+^ content, and higher electronic conductivity. The results showed that in the LNMOB0.01 sample, the Li^+^ diffusion coefficient was the highest and the charge-transfer resistance was the lowest. Yang et al. prepared typical pomegranate-like Ti-doped LiNi_0.4_Mn_1.6_O_4_ cathode materials (LNMTO) with no residue on the surface, which are anticipated to reduce the Li^+^ transmission path, weaken the electrochemical polarization, and improve the reaction kinetics (Figure 2g,h) [46]. Furthermore, during the electrochemical reaction, the strong Ti–O bond may stabilize the crystal structure by strengthening the lattice oxygen. Therefore, the LNMTO delivered a rapid charge–discharge performance with an obvious reversible capacity of 101 mAh g^−1^ between 3.5–5.0 V, which is observed even at 10 C and 84.4% capacity retention after 500 cycles at 1 C (Figure 2i). Doping causes the formation of doped element compounds on the LNMO surface, which necessarily reduces the doping efficiency and hinders the transmission of lithium ions during the lithiation-digestion process, thereby increasing the interface resistance and reducing the fast-charging ability. As a consequence, obtaining high-efficiency doping with heterogeneous components while avoiding the formation of surface residues remains a large challenge.

Similarly, a surface coating is another way to modify the fast-charging performance of LNMO. As an anode material for LIB, Nisar et al. used a large-scale ball milling technique to cover the LiNi_0.5_Mn_1.5_O_4_ surface with nano-sized zirconia (ZrO_2_) (Figure 3a) [41]. It was discovered that the as-prepared LiNi_0.5_Mn_1.5_O_4_ materials containing ZrO_2_ at 1.0 and 2.0 wt.% present delivered good rate performance and cycling life at room temperature. Moreover, the zirconia-modified LiNi_0.5_Mn_1.5_O_4_ materials were able to withstand current densities up to an 80 C rate. The as-prepared LiNi_0.5_Mn_1.5_O_4_ material with 1.0 wt.% ZrO_2_ exhibited a capacity of 94 mAh g^−1^ after 1200 cycles at a 40 C discharge rate, corresponding to 85.6% capacity retention at room temperature (Figure 3b). Furthermore, the as-prepared LiNi_0.5_Mn_1.5_O_4_ material coated with 2.0 wt.% ZrO_2_ possessed good electrochemical cycle life at 55 °C (corresponding to 76% capacity retention) as opposed to the ZrO_2_ material that had no coating. When exposed to 6 C fast charging and C/3 discharging for 300 cycles, the coated ZrO_2_ materials exerted outstanding cycle life (133 mAh g^−1^). LNMO materials were treated with distinct levels (0.5–2.0 wt.%) of silica (SiO_2_) on their surfaces utilizing affordable and large-scale ball mill technology, and the surface-modified materials displayed acceptable electrochemical characteristics when used with traditional liquid electrolytes (Figure 3c) [42]. By applying half-cell and full-cell tests, it was discovered that the coating improved the electrochemical characteristics at both room temperature and high temperature (25 °C and 55 °C). As shown by the TEM experiment carried out in situ, there was significant optimization of the coating and solid electrolyte interface characteristics, showing that the LNMO active particles and the SiO_2_ layer tightly adhered to each other and that the coating had strong wettability. In particular, the 1 wt.% SiO_2_-coated sample that underwent 400 cycles at 10 C, 40 C, and 80 C rates delivered good cycle life with the corresponding capacity retentions of 96.7, 87.9, and 82.4%. After being charged at 6 C for 500 cycles and drained at C/3 over those same cycles, the sample with 1 wt. percent SiO_2_-coating exerted excellent cycle performance (Figure 3d). Kuenzel et al. designed LNMO crystals with customized surface facets by carefully adjusting the synthesis parameters, providing excellent rate performance and improved interface stability (Figure 3e) [47]. The inclusion of barrier protection involved TMPO_x_ coatings optimized for the longer lifetime cycle, especially at high threshold potentials reaching up to 4.95 V, extreme temperatures of up to 40 °C, and high charge/discharge rates. Owing to the well-designed nature of the LNMO active material particles, LIBs adopting TMPO_x_-coated LMNO cathodes combined with Li_4_Ti_5_O_12_ anodes delivered good rate performance, with 80% of the minimal-rate performance and rapid rates of charge-discharge rates of 10 C together with excellent cyclic stability at an increased rate, as shown by the capacity retention of 95% after 1000 cycles (Figure 3f). The majority of these surface treatment technologies rely on sophisticated chemical processes, making it nearly impossible to regulate the uniformity and thickness of the coating. Under most circumstances, the interfacial resistance formed by the coating may help improve the material’s rate performance. Indeed, nano-sized thin coatings were beneficial to improve battery life and service life. Thick coatings, on the other hand, might serve as a movement impediment for lithium transference, resulting in a reduction in both capacity and rate characteristics. Therefore, quantitative research is required to reveal the trade-off between interface transfer/kinetics (rate performance and capacity) and interface protection (cycling stability) in a particular approach for battery coating.

Wei et al. prepared a surface-sulfided LiNi_0.5_Mn_1.5_O_4_ cathode material via the electrostatic interaction between sulfide ions with a negative charge and LiNi0.5Mn1.5O_4_ with positive charges (Figure 4a,b). The obvious gain of surface vulcanization is the production of a 3D permeable vulcanized layer, which is conducive to the formation of a stable cathode electrolyte interphase film that improves the diffusion kinetics of Li^+^ on the surface. The adsorption of SO_4_^2−^ on the surface of LiNi_0.5_Mn_1.5_O_4_ and the decrease in work function caused by surface sulfidation help to enhance the rate performance and cycle life of the LiNi_0.5_Mn_1.5_O_4_/electrolyte interface. Consequently, a discharge capacity of 93.4 mAh g^−1^ was delivered at 2 C after 2500 cycles, where the capacity retention was 74.9% and was significantly greater in contrast with the pristine electrode (45.3% after 1800 cycles) (Figure 4c). Gu et al. used C_24_H_20_BNa and C_24_H_20_PBr as precursors to synthesize high-voltage LNMO coated with boron co-doped carbon and phosphorus, and this coating was shown to effectively block the electrode-electrolyte interface side reaction (Figure 4d,e) [48]. In addition, the coating improves the weak conduction property of LNMO. When compared with the uncoated LNMO electrode, the P, B, and C-coated LNMO electrodes exhibited excellent rate performance as well as high cycle capacity. In particular, the P, B–C@LNMO-3 delivered high-capacity retention of 96.7% after 200 cycles at 1 C discharge rate and 111 mAh g^−1^ capacity at a 5 C discharge rate (Figure 4f). Several factors were identified as contributing to the optimized electrochemical performance of the P, B–C@LNMO materials, including their resistance to attack posed by the electrolyte and a small amount of water, increased electronic conduction, and increased Li^+^ ion transport. The effective combination of the surface coating and doping can effectively suppress the electrode-electrolyte interface side reaction and improve the low conductivity of LNMO. This is the most effective approach for modifying the LNMO to achieve the fast-charging performance of the LNMO cathode.

### 3.2. LiNi_x_Co_y_Mn_z_O_2_ (NCM)

A layered-structure high-nickel material, LiNi_x_Co_y_Mn_z_O_2_ (NCM), has a hypothetical specific capacity that exceeds 200 mA h g^−1^ and relatively low raw material cost. Therefore, it is used widely as a cathode in LIBs [22]. However, the NCM cathode material has certain shortcomings, including structure instability, mixing of cations, and rapid cycle life capacity decay that hinders its practical application in fast-charging LIBs. NCM has been reported to have a secondary spherical structure and is made of numerous smaller primary particles, which are arranged in a ring configuration. While the unique morphology of this cathode material has a larger contact area with the electrode, there are still problems such as particle cracking during the circulation process and oxygen generated by the side reaction of the electrolyte [50].

In 2018, Wu et al. developed the polyacrylonitrile sulfide (SPAN) as a safe and high-capacity anode in LIBs. They also introduced an elevated voltage cathode made of LiNi_1/3_Co_1/3_Mn_1/3_O_2_ (NCM-H) to a novel SPAN|NCM-H battery that is capable of performing quick charging operations (Figure 5a) [51]. These LIBs had outstanding cycling stability, as confirmed by their large capacity retention of 89.7% after 100 cycles at an elevated voltage of 3.5 V (that is, 4.6 V as opposed to Li^+^/Li). Furthermore, the excellent rate capacity was confirmed, and 78.7% of the initial capacity could still be delivered at 4.0 C. Additionally, 97.6% of the capacity could be charged within 2.0 C, which is far greater than the existing fast-charging application criteria of 80% (Figure 5b). The strong lithiation potential of SPAN (greater than 1.0 V vs. Li^+^/Li) increased security and prevented lithium deposition, while its elevated capacity of 640 mAh g^−1^ demonstrated a 43.5% greater energy density in contrast with the Li_4_Ti_5_O_12_-based batteries. Zhao et al. prepared a peanut-like multi-level nano/microstructure LiNi_0.5_Co_0.2_Mn_0.3_O_2_ cathode material utilizing a facile hydrothermal approach together with an elevated-temperature calcination method (Figure 5c) [52]. The parameterization findings indicated that H-NCM possesses features of greater lithium-ion diffusion factor, minimal resistance to charge transfer, larger specialized surface area, uniformly distributed particle sizes, smaller sizes of particles, and a higher degree of crystallinity, all of which are beneficial in the improvement of its electrochemical performance. Electrochemical tests revealed that the H-NCM material with a peanut-like shape had greater discharge capacities, with values of 176.5 and 159.9 mAh g^−1^ when receiving 0.2 and 10 C rates, respectively (Figure 5d). Their cycle life was significantly improved, exhibiting capacity retention of 90.0% even after 100 cycles at a 10 C rate (Figure 5e). The H-NCM cathodes delivered excellent performance suitable for commercial applications with an original discharge capacity at 1 C being shown as 2084.2 mA h g^−1^ and 92.17% capacity retention after 500 cycles. A model materials Li_1.2_Mn_0.54_Ni_0.13_Co_0.13_O_2_ (MNC) was generated by including a semi-hollow microsphere morphology and its surface reconfigured utilizing graphene/carbon nanotube dual layers and a surface-treated layer [25]. The unique structural configuration enabled a high tap density (2.1 g cm^−3^) and two-way ion diffusion channels. The dual surface coating, which was covalently bound to MNC through a C–O–M connection, significantly enhanced charge-transfer performance while also reducing electrode degradation. Due to the synergistic effect, the produced MNC cathode displayed good conformality and structural integrity over a long period of time, as well as a high volumetric energy density (2234 Wh L^−1^) and a capacitive behavior that was dominant. The constructed full battery with nano-graphite as the anode revealed a long cycling stability (1000 cycles), excellent rate property (70.3% retention capacity retention at 20 C), and an energy density of 526.5 Wh kg^−1^. A redox reaction involving Mn^7+^ and Co^2+^ was carried out by Tsai et al. to form cobalt manganese oxyhydroxide precipitates outside the Ni(OH)_2_, and the Ni^2+^ on the surface was partly oxidized to Ni^3+^ to facilitate the process (Figure 5f) [53]. CoSO_4_ was chosen as a precursor to avoid the presence of the self-limiting phenomenon. A result that was highly crystalline and free of impurities was achieved effectively. After calcination, a homogeneous distribution of components was achieved. The NMC-R surface morphology was ordered and demonstrated a small degree of cation mixing. It was discovered that the material synthesized by redox-assisted deposition exhibited a large starting capacity of 197 mAh g^−1^ and that the capacity retention was about 93% after 100 cycles (Figure 5g). The material also delivered 68 mAh g^−1^ greater capacity in contrast to that of the LiNiO_2_ materials at an elevated rate of 10 C (Figure 5h). Thus far, the electrochemical characteristics of the NCM811 cathode material were enhanced primarily through the surface coating and elemental doping. Nonetheless, structural optimization design is also a feasible improvement approach. In follow-up research, the safety performance of NCM cathode materials, cycle life, and storage performance was improved significantly, and the battery design was optimized to match the NCM cathode and electrolyte.

### 3.3. LiNi_x_Co_y_Al_z_O_2_ (NCA)

Layered LiNixCoyAlzO_2_ (NCA) is the most promising cathode material for LIBs due to its higher hypothetical capacity (278 mAh g^−1^) and excellent rate capacity [54,55]. Tesla Motors recently incorporated NCA materials in LIBs to drive its Model S, X, and 3 vehicles, which have an endurance mileage of 400–550 km on a single full charging session. The relative Ni concentration of the NCA cathode must be increased if the energy density of the cathode is to be improved [56]. When recharged to 4.3 V, a heavily delithiated NCA cathode tends to experience structural deterioration at the surface of the cathode particles, which is generated by an interaction between the volatile Ni^4+^ and the electrolyte, resulting in the formation of a harmful NiO-like rock salt impurity stage [57,58]. Microcracks are generated in Ni-rich layered cathodes (with a relative Ni percentage of less than 0.8) as a result of H_2_→H_3_ phase transitions (at a voltage of ≈4.15 V), which occur abruptly in the c-axis direction [59,60,61,62,63]. The aggressiveness of the microcracking escalates in direct proportion to the amount of Ni present [62]. The electrolyte is generated by the microcracks, which penetrates the internal setting of the particle and attacks the innermost primary particles, leading to progressive structural deterioration, which is correlated with a reduction in available capacity and ultimately catastrophic system malfunctions [56,57,60,61,62,63]. Furthermore, this deterioration results in the escape of oxygen from the host structure, which leads to a severe exothermal reaction that endangers the safety of the battery cells in certain circumstances. 

In 2020, Nie et al. used a Li_1.3_Al_0.3_Ti_1.7_(PO_4_)_3_ (LATP) as a rapid ion conductor powder for in-situ modification to design a LiNi_0.8_Co_0.15_Al_0.05_O_2_ (NCA) cathode material in LIBs that had a stable structure (Figure 6a) [64]. The application of NCA has been modified with 1.0 wt.% of LATP precursor, which may shield active materials from erosion caused by electrolytes and enhance structural stability throughout the charge-discharge process, thus exhibiting significant enhancements in the cycling and rate performance due to the synergistic impacts of the generated Ti^4+^ doping and LATP coating. Consequently, the coin battery constructed utilizing NCA-1.0 delivered an exceptional specific discharge capacity of 152.4 mAh g^−1^ as well as a ratio of capacity retention at 89.5% after 150 cycles at 2 C, excellent rate capacity of 153.2 mAh g^−1^ at 5 C, and an increased Li^+^ diffusion coefficient of 6.65 × 10^−10^ cm^2^ s^−1^ (Figure 6b). Nie et al. used nano-Al(OH)_3_ as the Al source with LiOH·H_2_O and Ni_0.84_Co_0.16_(OH)_2_ as precursors to co-calcin and synthesized layered Ni-rich materials with Li(Ni_0.84_Co_0.16_)_1−x_Al_x_O_2_ (0 < x < 0.1) (Figure 6c) [65]. It was discovered that when the amount of doped nano-Al (OH)_3_ powder reached 4%, its cationic disorder reduced by 1.69%, and the interfacial spacing of the (003) phase increased by 0.482 nm. The coin battery constructed with NC-A4 delivered a preliminary specific discharge capacity of 196.3 mAh g^−1^ and retained 170.7 mAh g^−1^ after 100 cycles at 1.0 C at 30 °C with an excellent capacity retention of 95.5% (Figure 6e). Furthermore, it may also deliver a good rate capacity of 162.1 mAh g^−1^ at a high current density of 5.0 C (Figure 6d). Moreover, the remarkable enhancement in the structural strength of NC-A4 as a result of trivalent Al-ions doping has been attributed to the optimization in its electrochemical characteristics, which could also maintain steady crystalline structures even during lithiation/delithiation mechanisms and ameliorate the increment in polarization extent and Rct value. When Li[Ni_0.855_Co_0.13_Al_0.015_]O_2_ (NCA85) was doped with Nb, it changed the main particle shape, allowing for more exact control over the microstructure of the material (Figure 6f) [26]. The addition of the Nb dopant (1 mol percent) led the primary particles to become longer and more oriented in the radial direction, resulting in a structure that effectively dissipates the sudden internal strain induced by H_2_→H_3_ phase transitions towards the charge end. The elimination of internal strain enhanced the protracted cycling stability achieved by the Nb-doped NCA85 cathode, which retained 90% of its original capacity after 1000 cycles and retained 57.3% after the same number of cycles compared to the undoped cathode (Figure 6g). Additional advantages of the Nb-doped NCA85 cathode include enhanced mechano-chemical stability and the ability to charge more rapidly. Thus, even it was charged at 3 C, the Nb-doped NCA85 cathode was able to cycle for 500 cycles without losing its stability (20 min was required to achieve full charge) (Figure 6h). Combining morphology control with a surface coating, composite structure, and ion doping may be a more effective strategy for an in-depth study of the structure and performance of NCA cathode materials, which is helpful in the rational design and structural optimization of NCA cathode materials. 

### 3.4. Li-Rich Mn-Based (LRM)

As the demand for high energy density continues to increase, lithium-rich manganese-based materials have attracted increasing attention due to their elevated voltage (>4.5 V) and greater capacity (>250 mAh/g) [66,67]. LRM materials can be described as Li_1+(x/(2+x))_Mn_2x/(2+x)_M_−2+(6/(2+x))_O_2_ (M′ = Mn + M) or xLi_2_M_n_O_3_·(1−x)LiMO_2_ (M is a 3d or 4d transition metal) [67]. For the first time, Thackeray et al. synthesized Li_1.09_Mn_0.91_O_2_ by calcining Li_0.36_Mn_0.91_O_2_ and LiI [68,69,70]. Layered Li_2_MnO_3_⋅LiCoO_2_ solid solution material was introduced in 1997 by Numata et al. [71]. They prepared Li(Li_x/3_Mn_2x/3_Co_1−x_)O_2_⋅yLi_2_O via calcination at relatively elevated temperatures. The design idea of a solid solution was subsequently applied in the preparation of a wide variety of different electrode materials [20,72,73,74]. Lithium-rich manganese-based materials have a special morphology and distinct electrochemical properties. These materials are widely regarded as cathode materials for next-generation LIBs because of their advantages such as high voltage and high capacity. Notably, many concerns and hurdles remain, when it comes to explaining and resolving their degradable electrochemical performance, which should be addressed. He et al. enhanced the electrochemical performance and ionic conductivity of the LRM cathode by adopting the distinct distribution habits of La^3+^ and Zr^4+^ (Figure 7a) [75]. In the presence of lattice mismatch, Zr atoms have a higher likelihood of accumulating on the surface of the particles, while La atoms have a higher likelihood of doping into the bulk, as demonstrated by theoretical analysis and experimental observations. Consequently, the surface is transformed by the island-shaped, high conductivity materials, where the created built-in electrostatic force speeds up Li-ion diffusion and promotes the stability of the solid-liquid interface. Moreover, the large-scale doping of strong Zr–O bonds widens the Li-layer while simultaneously improving the layered framework. It was discovered that the modified LMR cathode had significantly reduced Li-ion diffusion coefficients, allowing it to retain a higher specific capacity of 192.6 mA h g^−1^ after 200 cycles at 2 C (Figure 7b). LMR Li_1.2_Mn_0.6_Ni_0.2_O_2_ was modified via phosphorus (P)-doping to increase the Li^+^ conductivity in the bulk material (Figure 7c) [76]. This was accomplished by widening the lithium interlayer spacing, increasing electron transfer, and strengthening structural stability, leading to an increase in rate and security property. It was discovered that the addition of P^5+^ strongly enhanced the spacing seen between (003) crystal planes from about 0.47 to 0.48 nanometers and that this enhanced structural stability was caused by the formation of strong covalent bonds with oxygen atoms, resulting in improved rate property (capacity retention considerably increased from 38 to 50 percent at 0.05 to 5 C, respectively) and thermal stability (increase by 50 percent heat release when contrasted with pristine material) (Figure 7d). Calculations based on first-principles evidence revealed that P-doping facilitates the flow of excited electrons from the valence to the conduction bands. P has the ability to establish a strong covalent bond, contributing to the stabilization of the structure of the material. Moreover, the P^5+^-doped LMR altered with a solid-state electrolyte displayed superior cycling properties for up to 200 cycles with 90.5% capacity retention and improved the ICE from 68.5% (pristine) or 81.7% (P-doped LMR) to 88.7% (Figure 7e). Before its commercial applications, more efforts should be focused on the charging as well as the discharging mechanism. In particular, the discharging and charging mechanism improves the preliminary Coulomb efficiency and eliminates the voltage attenuation in the course of the cycle. Moreover, synthetic methods appropriate for large-scale manufacture need to be thoroughly explored. From the point of view of energy density, cost, and cycling stability, LRM is the future direction as the most promising fast charging cathode material.

Moreover, the content of the Li element in the layered oxide plays an important role in the performance of Li-ion batteries [77]. However, at higher calcination temperatures, the element Li tends to volatilize. This effect should be mentioned, which is crucial in the preparation of Li-based layered oxides for LIBs. In addition to morphology, elemental composition, surface properties, and complex structure, the electronic structure (valence, spin, and covalent states) and local structure (bond lengths and bond angles) of lithium base-like oxides are also important and can be extracted precisely from soft and hard XAS spectra [78]. These issues should be a cause for concern.

## 4. Conclusions and Outlook

The unique structure of layered-oxide materials has attracted increasing attention as one of the most prospective fast-charging cathodes for state-of-the-art LIBs. This article summarized the recent progress in layered fast-charging cathode materials and their component improvement, morphology control, ion doping, surface coating, and composite structure, as shown in Table 1. However, there has been little research on the cathodes of fast-charging LIBs. Further, the lack of precise crystal models and the complex reaction mechanisms of layered oxides have caused many unresolved problems and hindered their practical applications. The following challenges need to be overcome in order to obtain high-performance fast-charging cathodes and apply layered materials in fast-charging LIBs.

1. The first challenge is to inhibit the irreversible redox of anions, stabilize the electrolyte-electrode interface reaction, prevent electrolyte degradation, and suppress parasitic reactions. Expended oxygen caused by nonreversible anion activity is the main cause for the decrease in the stability of the interface, which requires further treatment, particularly under rapid-charging conditions. This is a fundamental demand for security and good cycling stability.

2. The second issue is to boost the ionic and electronic conduction ability of layered oxides while retaining their electrochemical stability. The low-rate property and slow diffusion kinetics have always been the primary problems in industrial applications owing to diminished ionic and electronic conductivity.

3. The ratio of the cation/anion redox reaction is optimized to obtain the maximum capacity contribution. Cationic redox reactions based on layered oxides result in poor energy density and specific capacity. Conversely, increased anionic redox activities often lead to the slowing of structural decomposition and reaction kinetics in the process of cycling. Consequently, balancing the contributions of cations and anions is essential for the high performance and good cycling stability of layered oxides.

4. Retaining the integrity of the layered-oxide crystal structure without sacrificing its original merits is important. The addition of foreign components (such as metal doping and surface coatings) must not destroy the uniformity of the structural layer and needs to maintain its intrinsic merits, including fast charging and high specific capacity.

Other practical issues should also be considered. Most layered oxides have mainly been studied as half-cells and evaluating their true electrochemical performance in practical applications is difficult. Therefore, more research efforts on the whole battery are needed, such as selecting suitable anode materials and developing effective fast-charging electrolytes. The safety issues of layered oxides, thermal runaway, gas escape, and other possible side reactions are key factors that need to be explored further. In addition to optimizing its performance, battery design optimization (including more reasonable cathode material matching, security additives, thermal conductivity design, and size design), and module and package design optimizations (such as battery protection design, safety protection, and heat dissipation design) are additional requirements for the large-scale application of fast-charging LIBs on the basis of layered oxides. Furthermore, there is a shortage of lithium, nickel, and cobalt resources, and they are expensive. Therefore, cathode materials that are both cobalt- and nickel-free and lithium-rich are needed, as are battery recycling technologies. Based on the huge potential of layered oxides, the commercialization of layered oxides can be realized soon, and the energy density of the next-generation of layered-oxide systems can be raised to a new level.

## Figures and Tables

**Figure 1 molecules-28-04007-f001:**
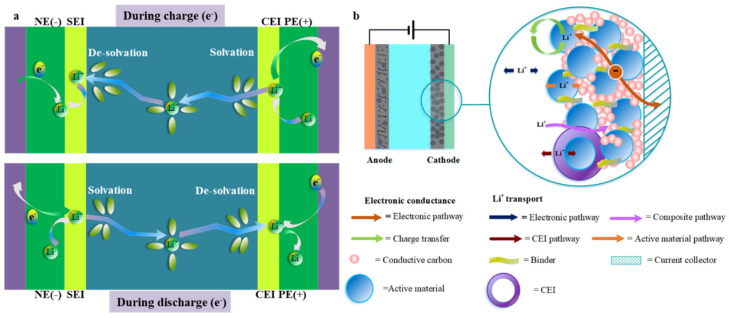
(**a**) Schematic diagram of lithium-ion transfer during charging and discharging, (**b**) The main kinetic processes that occur on the cathode.

**Figure 2 molecules-28-04007-f002:**
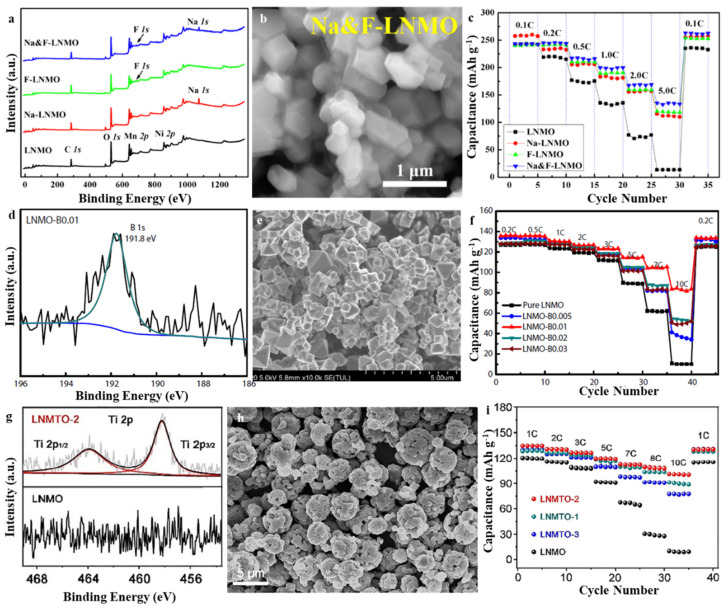
(**a**) XPS of LNMO, Na-LNMO, F-LNMO, and Na and F-LNMO, (**b**) SEM of Na and F-LNMO, (**c**) Rate performance of LNMO, Na-LNMO, F-LNMO, and Na and F-LNMO. Reproduced with permission from [41]. Copyright 2017, Elsevier. (**d**) XPS of LNMO-B0.01, (**e**) SEM of LNMO-B0.01, (**f**) Rate performance of LNMO and the LNMO with various B content. Reproduced with permission from [45]. Copyright 2021, Elsevier. (**g**) XPS of LNMTO-2, (**h**) SEM of LNMTO-2, (**i**) Rate performance of LNMO and LNMTO with various Ti content. Reproduced with permission from [46]. Copyright 2021, Elsevier.

**Figure 3 molecules-28-04007-f003:**
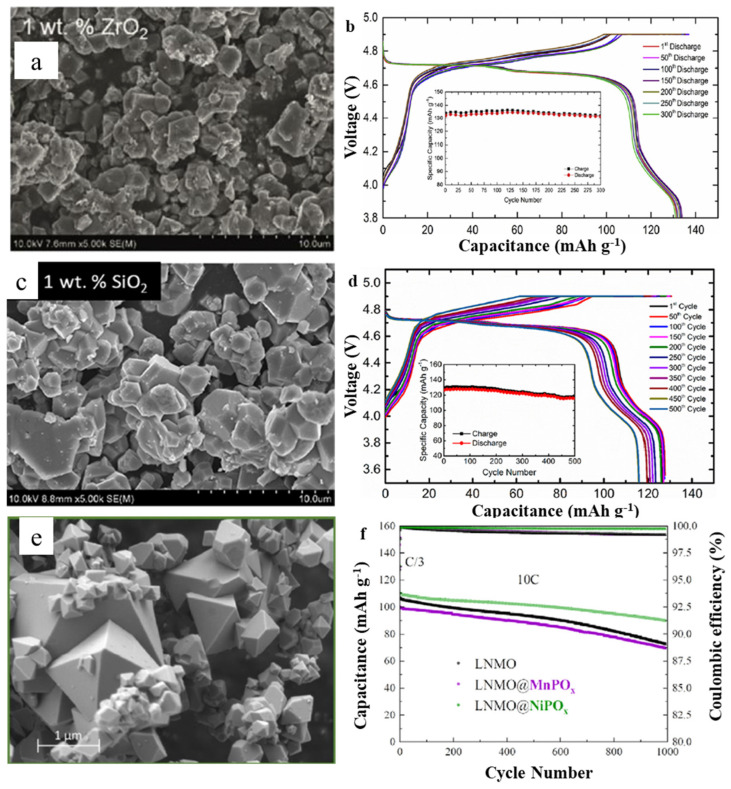
(**a**) SEM of 1.0 wt.% ZrO_2_-coated LNMO, (**b**) Charge/discharge profile of 1.0 wt.% ZrO_2_-coated LNMO charged at a 6 C rate and discharged at a C/3 rate. Reproduced with permission from [41]. Copyright 2018, Elsevier. (**c**) SEM of 1 wt.% SiO_2_ coated LNMO, (**d**) Charge/discharge profile of 1 wt.% SiO_2_-coated LNMO charged at a 6 C rate and discharged at a C/3 rate. Reproduced with permission from [42]. Copyright 2019, American Chemical Society. (**e**) SEM of LNMO@NiPO_x_, (**f**) Long-term constant current cycling at 10 C for 1000 cycles with LNMO, LNMO@MnPO_x_, and LNMO@NiPO_x_. Reproduced with permission from [47]. Copyright 2020, Elsevier.

**Figure 4 molecules-28-04007-f004:**
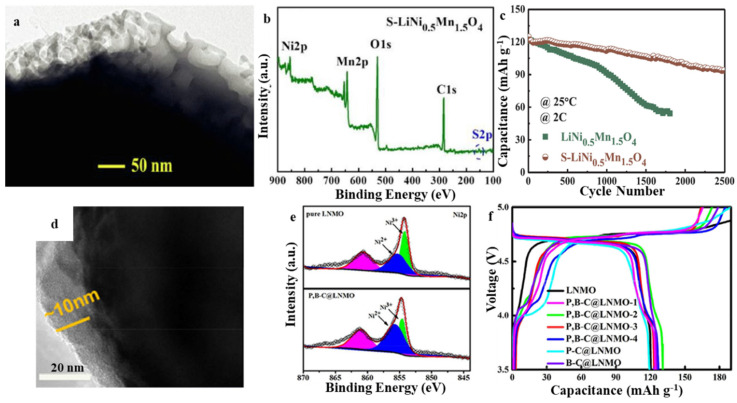
(**a**) TEM of surface-sulfidized LiNi_0.5_Mn_1.5_O_4_, (**b**) XPS of surface-sulfidized LiNi_0.5_Mn_1.5_O_4_, (**c**) Cycling performances of the pristine and surface-sulfidized LiNi_0.5_Mn_1.5_O_4_. Reproduced with permission from [49]. Copyright 2020, Elsevier. (**d**) TEM of P, B–C@LNMO-3, (**e**) XPS of P, B–C@LNMO-3, (**f**) Charge/discharge profile of LNMO and carbon coated LNMO with different coating amounts. Reproduced with permission from [48]. Copyright 2019, Wiley.

**Figure 5 molecules-28-04007-f005:**
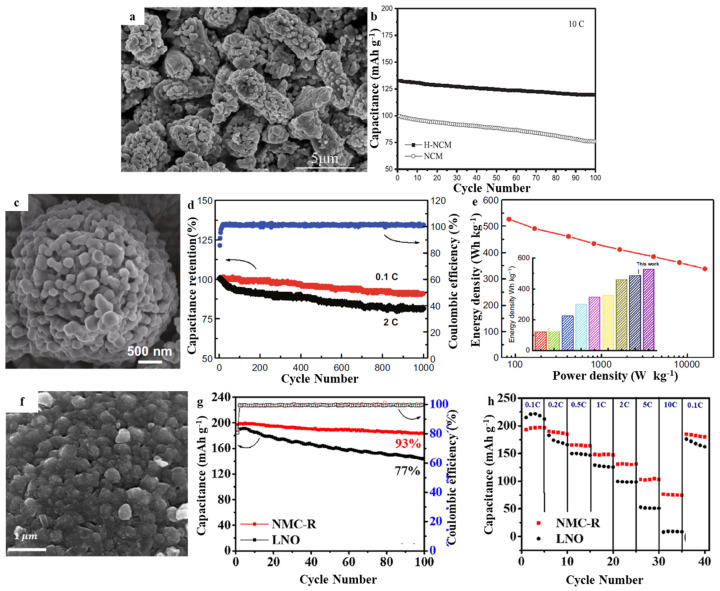
(**a**) SEM of H-NCM, (**b**) Cycling performance of H-NCM and NCM at 10 C rate. Reproduced with permission from [51]. Copyright 2019, Wiley. (**c**) SEM of MNC, (**d**) Cycling performance of assembled full cell at 0.1 C and 2 C, (**e**) Ragone plots based on total mass of cathode and anode. Reproduced with permission from [52]. Copyright 2020, Elsevier. (**f**) SEM of NMC-R, (**g**) Cycling performance of NMC-R and LNO at 0.5 C, (**h**) Rate discharge capability of NMC-R and LNO at different rate. Reproduced with permission from [53]. Copyright 2021, Springer.

**Figure 6 molecules-28-04007-f006:**
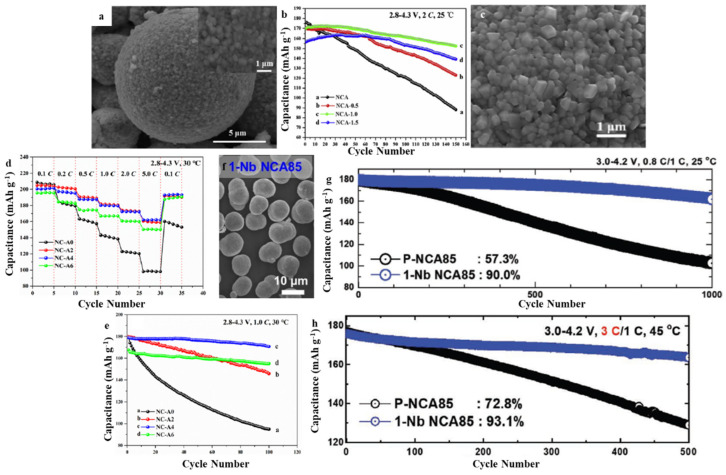
(**a**) SEM of NCA-1.0, (**b**) Rate performance of NCA, NCA-0.5, NCA-1.0, and NCA-1.5. Reproduced with permission from [64]. Copyright 2020, Elsevier. (**c**) SEM of NC-A4, (**d**) Rate, (**e**) Cycling, performance of NC-A0, NC-A2, NC-A4, and NC-A6. Reproduced with permission from [63]. Copyright 2021, Elsevier. (**f**) SEM of 1-Nb NCA85, (**g**) Cycling performance at 1.0 C in full cells, (**h**) Cycling performance at 3 C in full cells. Reproduced with permission from [26]. Copyright 2021, Wiley.

**Figure 7 molecules-28-04007-f007:**
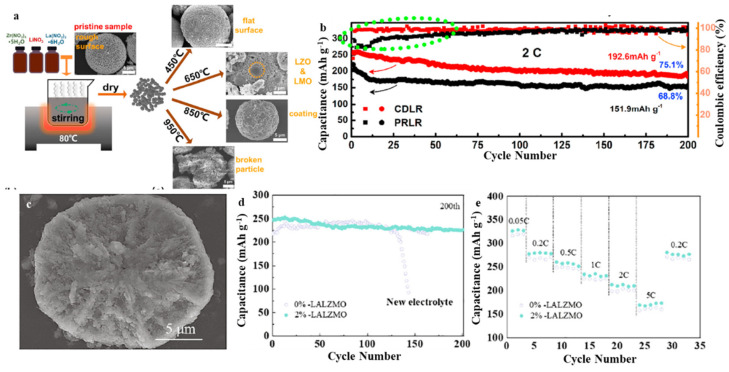
(**a**) Schematic illustration for m-LRM synthesized at different temperatures, (**b**) Cycling performance of PRLR and CDLR electrodes at 2 C. Reproduced with permission from [75]. Copyright 2021, Elsevier. (**c**) SEM of Li_1.1_P_0.02_Ni_0.2_Mn_0.6_O_2_, (**d**) Cycling performance, (**e**) Rate performance of LALZMO-coated LPMNO with coating ratios of 0 wt.% and 2 wt.%. Reproduced with permission from [76]. Copyright 2021, Elsevier.

**Table 1 molecules-28-04007-t001:** Comparison of the reported fast charging LIBs based on layered-oxide cathode materials.

Layered-Oxide Materials	Voltage Window (V)	Maximum Capacity (mAh g^−1^)	Capacity@High c-Rate (mAh g^−1^)	Capacity Retention	Ref.
Co-doped Na and F-LNMO	2.0–4.8	245 (0.2 C)	167 (5 C)	100% after 100 cycles (0.2 C)	[44]
LNMO-B0.01	3.5–5.0	136.1 (0.2 C)	82.9 (10 C)	83% after 500 cycles (3 C)	[45]
LNMTO	3.5–5.0	134 (1 C)	101 (10 C)	84.4% after 500 cycles (1 C)	[46]
ZrO_2_-Modified LNMO	4.0–4.8	133 (0.5 C)	101 (80 C)	85.6% after 1200 cycles (40 C)	[41]
SiO_2_-Modified LNMO	3.4–5.0	132 (0.5 C)	105 (80 C)	96.7% after 400 cycles (10 C)	[42]
LNMO@TMPOx	1.5–4.8	128 (0.1 C)	110 (10 C)	82% after 1000 cycles (10 C)	[47]
S-LNMO	3.0–4.8	125 (0.5 C)	116 (6 C)	74.9% after 2500 cycles (2 C)	[44]
P, B-C@LNMO	3.5–5.0	130.7 (0.1 C)	111 (5 C)	96.7% after 200 cycles (5 C)	[48]
NCM-H	0.5–3.5	175 (0.1 C)	-	89.7% after 100 cycles (0.5 C)	[51]
Peanut-like H-NCM	3.0–4.3	176.5 (0.2 C)	159.9 (10 C)	90% after 100 cycles (10 C)	[52]
MNC	2.5–4.8	276 (0.1 C)	189 (10 C)	70.3% after 1000 cycles (20 C)	[25]
NMC-R	2.4–4.3	197 (0.1 C)	68 (10 C)	93% after 100 cycles (0.5 C)	[53]
NCA	2.8–4.3	198.5 (0.1 C)	153.2 (0.5 C)	89.5% after 1500 cycles (2 C)	[64]
NCA	2.8–4.3	196.3 (1 C)	162.1 (5 C)	95.5% after 100 cycles (1 C)	[65]
NCA85	2.7–4.3	-	-	90% after 1000 cycles (1 C)	[26]
PRLR	2.0–4.8	268 (0.1 C)	150 (5 C)	75.1% after 200 cycles (2 C)	[75]
P^5+^-LMR	2.0–4.7	317 (0.05 C)	175 (5 C)	90.5% after 200 cycles (5 C)	[76]

Acronym definitions: Sodium/fluorine co-doped Li_1_._2_Ni_0_._2_Mn_0_._6_O_2_ (Na and F-LNMO), Boron-doped LiNi_0.495_B_0.01_Mn_1.495_O_4_ (LNMO-B0.01), Ti-doped LiNi_0.4_Mn_1.6_O_4_ (LNMTO), LiNi_0.5_Mn_1.5_O_4_ (LNMO), LiNi_1/3_Co_1/3_Mn_1/3_O_2_ (NCM-H), LiNi_0.5_Co_0.2_Mn_0.3_O_2_ (H-NCM), Li_1.2_Mn_0.54_Ni_0.13_Co_0.13_O_2_ (MNC), LiNi_0.96_Co_0.03_Mn_0.01_O_2_ (NMC-R), LiNi_0.8_Co_0.15_Al_0.05_O_2_ (NCA), Li(Ni_0.84_Co_0.16_)_1−x_Al_x_O_2_ (NCA), Li[Ni_0.855_Co_0.13_Al_0.015_]O_2_ (NCA85), Li_1.2_Ni_0.13_Co_0.13_Mn_0.54_O_2_ (PRLR), Li_1.2_Mn_0.6_Ni_0.2_O_2_ (LMR).

## Data Availability

Not applicable.

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
