# Peer review of "Layered-Oxide Cathode Materials for Fast-Charging Lithium-Ion Batteries: A Review"

_molecules, 2023, doi:10.3390/molecules28104007_

Round 1

Reviewer 1 Report

The authors reviewed the progress of layered-oxide cathode materials for fast-charging lithium-ion batteries. Different kinds of layered-oxide cathode materials have been reviewed, including spinel LiNi0.5Mn1.5O4, LiNixCoyMnzO2, LiNixCoyAlzO2, etc. The review here is useful for the authors in the research field. I only have three minor comments for the review.

1) The performance of these layered-oxide cathode materials seems to be based on coin cell evaluation in this review paper. The authors should add more processes on the lithium batteries in pouch cells.   

2) The authors may add one or several paragraphs to discuss the different synthesis methods of these layered-oxide cathode materials.

3) From the point of view of energy density, cost, cycling stability, what are the advantage and disadvantages of these different layered-oxide materials?

3) Several related references in this area should be cited, Nature Communications 14.1 (2023): 146; Angewandte Chemie International Edition 59 (34), 14313-14320

Minor editing of English language required

Author Response

Thank you for your letter and for the reviewers’ useful comments and suggestions on our manuscript entitled “Layered-oxide cathode materials for fast-charging lithium-ion batteries: A review” (ID: molecules-2372756). These comments are very valuable and helpful for improving our paper, as well as the important guiding significance to our manuscript. We have studied comments carefully and have made correction which we hope meet with approval. The point to point responds to these comments and suggestions are listed as following (the major revisions are shown in yellow background in the manuscript):

REVIEWER REPORT(S):

Referee: 1

Comments to the Author

The authors reviewed the progress of layered-oxide cathode materials for fast-charging lithium-ion batteries. Different kinds of layered-oxide cathode materials have been reviewed, including spinel LiNi0.5Mn1.5O4, LiNixCoyMnzO2, LiNixCoyAlzO2, etc. The review here is useful for the authors in the research field. I only have three minor comments for the review.

Response: Thanks for the reviewer’s highly positive remarks and valuable guidance.

  1. The performance of these layered-oxide cathode materials seems to be based on coin cell evaluation in this review paper. The authors should add more processes on the lithium batteries in pouch cells.

Response: Thank you for your valuable advice. We strongly agree with the reviewers and we have rediscovered the relevant literature. However, most of the current fast charging performance of layered oxide cathode materials has been evaluated based on coin cells. We hope the reviewers will understand.

  1. The authors may add one or several paragraphs to discuss the different synthesis methods of these layered-oxide cathode materials.

Response: We highly appreciate your comments. We strongly agree with the reviewers. In fact, we present the fast charging performance of each of the four layered-oxide cathode materials (LiNi0.5Mn1.5O4, LiNixCoyMnzO2, LiNixCoyAlzO2 and Li-rich Mn-based (LRM)) modifications. These four methods of synthesis of layered oxides are actually discussed separately in the introduction of their fast charging performance. Therefore, we have not carried out a discussion on the different methods of layer oxide synthesis. We hope the reviewers will understand.

  1. From the point of view of energy density, cost, cycling stability, what are the advantage and disadvantages of these different layered-oxide materials?

Response: Many thanks for kind recommendation. From the point of view of energy density, cost, cycling stability, materials containing cobalt (LiNixCoyMnzO2 and LiNixCoyAlzO2) may not be cost advantageous. In contrast, Li-rich Mn-based (LRM) materials are more advantageous for fast charging.

  1. Several related references in this area should be cited, Nature Communications 14.1 (2023): 146; Angewandte Chemie International Edition 59 (34), 14313-14320.

Response: Thanks for the reviewer’s kind suggestion. We have cited these references and made yellow background mark in our revised manuscript. As following:

  1. Liang, J.; Zhu, Y.; Li, X.; Luo, J.; Deng, S.; Zhao, Y.; Sun, Y.; Wu, D.; Hu, Y.; Li, W.; Sham, T. K.; Li, R.; Gu M.; Sun, X. A gradient oxy-thiophosphate-coated Ni-rich layered oxide cathode for stable all-solid-state Li-ion batteries, Nat. Commun., 2023, 14, 1461.
  2. Xiao, B.; Liu, H.; Chen, N.; Banis, M. N.; Yu, H.; Liang, J.; Sun, Q.; Sham, T. K.; Li, R.; Cai, M.; Botton, G. A.; Sun, X. Size-Mediated Recurring Spinel Sub-nanodomains in Li- and Mn-Rich Layered Cathode Materials, Chem. Int. Ed., 2020, 59, 14313–14320.

Once again, thank you very much for your comments and suggestions.

Thank you the reviewers for the kind advices. If there have any problems or questions about our manuscript, please do not hesitate to let us know.

Sincerely yours,

Le Li

Reviewer 2 Report

In this review, the authors well summarize and discuss the different strategies applied in the layered-oxide cathode for fast-charging lithium-ion batteries, where related future development directions are also pointed out. Overall, this focus is timely and the discussions are systematic. Therefore, it can be accepted after addressing my comments as follows, which will help improve the quality of this version.

1.     In the suitable figures, the crystal structures of layered oxides should be given for better understanding. Also, necessary discussions about the structures should be provided.

2.     The content of Li element in layered oxides exerts an important role in the performance of lithium-ion batteries. However, Li element tends to volatilize under high calcination temperatures. This effect should be mentioned, which is critical in the preparation of Li-based layered oxides for lithium-ion batteries. Please refer to the paper with DOI of 10.1016/j.jechem.2023.03.033.

3.     Besides the morphology, element component, surface property, and composite structure of Li-based layered oxides, their electronic structure (valence state, spin state, and covalence) and local structure (bond length and bond angle) are also important, which can be precisely extracted from the soft and hard XAS spectra. Brief but necessary discussions should be given and some literatures can be referred to (DOI: 10.1063/5.0083059).

4.     For the future development directions, the authors are suggested to point out the development of activity descriptors on Li-based layered oxides for lithium-ion batteries, which will help save time and cost in terms of material design.

Minor editing of English language required.

Author Response

Thank you for your letter and for the reviewers’ useful comments and suggestions on our manuscript entitled “Layered-oxide cathode materials for fast-charging lithium-ion batteries: A review” (ID: molecules-2372756). These comments are very valuable and helpful for improving our paper, as well as the important guiding significance to our manuscript. We have studied comments carefully and have made correction which we hope meet with approval. The point to point responds to these comments and suggestions are listed as following (the major revisions are shown in yellow background in the manuscript):

REVIEWER REPORT(S):

Referee: 2

Comments to the Author

In this review, the authors well summarize and discuss the different strategies applied in the layered-oxide cathode for fast-charging lithium-ion batteries, where related future development directions are also pointed out. Overall, this focus is timely and the discussions are systematic. Therefore, it can be accepted after addressing my comments as follows, which will help improve the quality of this version.

Response: Thanks for the reviewer’s kind suggestions. All the suggestions are very important

and having significant guidance for this work.

  1. In the suitable figures, the crystal structures of layered oxides should be given for better understanding. Also, necessary discussions about the structures should be provided.

Response: Thank you for your valuable advice. We strongly agree with you that the crystal structure of the layered oxide should be provided. However, the crystal structures of the four layered-oxide cathode materials (LiNi0.5Mn1.5O4, LiNixCoyMnzO2, LiNixCoyAlzO2 and Li-rich Mn-based (LRM)) discussed in our manuscript are more complicated. Therefore, we just did not provide it. We hope the reviewers will understand.

  1. The content of Li element in layered oxides exerts an important role in the performance of lithium-ion batteries. However, Li element tends to volatilize under high calcination temperatures. This effect should be mentioned, which is critical in the preparation of Li-based layered oxides for lithium-ion batteries. Please refer to the paper with DOI of 10.1016/j.jechem.2023.03.033.

Response: Many thanks for kind recommendation. Based on the reviewers' comments, we have added these discussions to our revised manuscript and made yellow background mark in our revised manuscript.

  1. Guan, D.; Shi, C.; Xu, H.; Gu, Y.; Zhong, J.; Sha, Y.; Hu, Z.; Ni, M.; Shao, Simultaneously mastering operando strain and reconstruction effects via phase-segregation strategy for enhanced oxygen-evolving electrocatalysis. J. Energy Chem., 2023, DOI: 10.1016/j.jechem.2023.03.033.

  1. Besides the morphology, element component, surface property, and composite structure of Li-based layered oxides, their electronic structure (valence state, spin state, and covalence) and local structure (bond length and bond angle) are also important, which can be precisely extracted from the soft and hard XAS spectra. Brief but necessary discussions should be given and some literatures can be referred to (DOI: 10.1063/5.0083059).

Response: Thanks for the reviewer’s kind suggestions. Accordingly to the reviewers' comments, we have added these discussions to our revised manuscript and made yellow background mark in our revised manuscript.

  1. Guan, D.; Zhong; J.; Xu, H.; Huang; Y. C.; Hu; Z.; Chen; B.; Zhang; Y.; Ni; M.; Xu; X.; Zhou; W.; Shao, Z. A universal chemical-induced tensile strain tuning strategy to boost oxygen-evolving electrocatalysis on perovskite oxides. Appl. Phys. Rev., 2022, 9, 011422.

  1. For the future development directions, the authors are suggested to point out the development of activity descriptors on Li-based layered oxides for lithium-ion batteries, which will help save time and cost in terms of material design.

Response: Thank you for your valuable advice. According to the reviewers' comments, we have added these discussions to our revised manuscript and made yellow background mark in our revised manuscript.

Once again, thank you very much for your comments and suggestions.

Thank you the reviewers for the kind advices. If there have any problems or questions about our manuscript, please do not hesitate to let us know.

Sincerely yours,

Le Li

Round 2

Reviewer 1 Report

Can be accepted.

Reviewer 2 Report

The revised version can be published.